# Exposure to self-reported traumatic events and probable PTSD in a national sample of Poles: Why does Poland's PTSD prevalence differ from other national estimates?

**Marcin Rzeszutek**[1]*, **Małgorzata Dragan**[1], **Maja Lis-Turlejska**[2], **Katarzyna Schier**[1], **Paweł Holas**[1], **Katarzyna Drabarek**[1], **Angelika Van Hoy**[1], **Małgorzata Pięta**[1], **Cecylia Poncyliusz**[1], **Magdalena Michałowska**[1], **Gabriela Wdowczyk**[1], **Natalia Borowska**[1], **Szymon Szumiał**[1]

**1** Faculty of Psychology, University of Warsaw, Warsaw, Poland, **2** Faculty of Psychology, SWPS University of Social Sciences and Humanities, Warsaw, Poland

* marcin.rzeszutek@psych.uw.edu.pl

## Abstract

### Background

There is a lack of studies on trauma exposure and PTSD prevalence in Poland on representative samples. Available data from studies on convenient samples show very high rates of probable PTSD compared with relevant estimates in other countries.

### Objective

This study aimed to measure the exposure to self-report traumatic events (PTEs) and to estimate the current rate of prevalence of probable posttraumatic stress disorder (PTSD) in accordance with DSM–5 criteria in a population-based sample of Poles. Additionally, the link between PTSD intensity and level of life satisfaction was investigated.

### Method

A representative sample of 1,598 adult Poles was recruited. Probable PTSD was assessed with the Posttraumatic Diagnostic Scale for DSM–5 (PDS–5) and the Satisfaction with Life Scale (SWLS) was also used.

### Results

The findings showed that 60.3% of Poles had experienced at least one PTE and 31.1% of those who had been exposed to trauma reported symptoms of PTSD. At the level of the entire sample, the obtained rate for probable PTSD was 18.8%. The traumatic events with the highest probabilities of PTSD symptoms were child abuse and sexual assault. Levels of life satisfaction were significantly lower in the group of participants with probable PTSD.

**Data Availability Statement:** All relevant data are within the manuscript and its Supporting Information files.

**Funding:** This project has received funding from the Ministry of Science and Higher Education in Poland (Ministerstwo Edukacji i Nauki), Award Number: 501-D125-20-0004316. The funders had no role in study design, data collection and analysis, decision to publish, or preparation of the manuscript.

**Competing interests:** The authors have declared that no competing interests exist.

## Conclusions

We found that the current prevalence of probable PTSD in Poland is intriguingly high relative to rates reported in comparable representative samples from other countries across the world. Possible mechanisms are discussed, including a lack of social acknowledgement of WWII and other traumas as well as poor access to trauma-focused care. We hope that this research may inspire more studies investigating cross-national differences in PTSD and trauma exposure.

## Introduction

Recent studies have shown posttraumatic stress disorder (PTSD), a common and highly debilitating mental disorder, is observed in nearly all countries [1–3]. While a huge amount of research has been devoted to individual-level risk factors for PTSD, including versatile pre-trauma, peritrauma, and posttrauma variables [for meta-analyses, see, e.g., [4–7]], only a few studies have explored country-level predictors of this disorder and their backgrounds (e.g., [8]). As such, researchers may observe considerable and–to a large extent–unexplained variance in PTSD rates across countries, which may be related to both methodological factors (e.g., sampling or measurement design) and the socioeconomic, institutional, or historical characteristics of each country [1, 9, 10]. For example, US national epidemiological studies have shown that while approximately 83–89% of Americans experienced at least one potentially traumatic event (PTE) in the course of their lives, and in the past 6–12 months, PTSD prevalence among Americans ranged from 3.8% to 4.7% [11–13]. Over 64% of the population of Europe experienced at least one PTE, with PTSD diagnosis (in the past 12 months) ranging from 0.4% 0.5% (in Spain and Germany, respectively) to from 2.0% to 3.8% in pre-war Ukraine and Northern Ireland, respectively [14–16]. Croatia, with a 6.67% rate of PTSD, was identified as an outlier country, having been strongly affected by the Yugoslav wars [17]. The available epidemiological studies from other continents showed rather similar patterns of trauma exposure and current PTSD prevalence–respectively, 57% and 1.3% in Australia [18], 78.8% and 1.5% in South Korea [19], 68.3% and 6.5% in Colombia [20], and 78.8% and 3.5% in South Africa [21]. The prevalence rates of experiencing at least one potentially traumatic event and probable PTSD obtained in research from various countries are summarized in Table 1. These studies rely on self-report and follow the DSM-IV PTSD diagnosis.

Research conducted to date in Poland has resulted in significantly different estimates. Some studies conducted among Poles exposed to at least one traumatic event showed that PTSD prevalence ranged from approximately 20–30% in convenience samples taken from the general population [e.g., 22, 23] to nearly 35–89% in clinical samples [24–28]. A recent meta-analysis of research evaluating the prevalence of probable PTSD in Poland showed a high overall rate of this disorder, with estimates ranging from 22% to 41% [29]. Similar estimates were found in those who survived World War II (WWII), which ranged from 29.4% to 38.3% [25, 26].

However, previous studies on PTSD prevalence in Poland had a range of limitations. They were done on convenient, not representative samples. Moreover, no study has analyzed associations between specific types of traumatic events and the intensity of PTSD symptoms. The types and intensities of traumas may allow the evaluation of whether some of these events were underrepresented or overrepresented in the data structure included in the meta-analysis by Szumiał [29].

**Table 1. Prevalence rates of experiencing at least one potentially traumatic event and probable PTSD in various countries.**

| Country | Total with any PTE | PTSD rate % |
|---|---|---|
| Spain | 54.0 | 0.4 |
| France | 72.7 | 1.4 |
| Northern Ireland | 60.6 | 3.8 |
| Netherlands | 65.6 | 1.2 |
| Belgium | 65.8 | 0.6 |
| Germany | 67.3 | 0.5 |
| Ukraine | 84.6 | 2.0 |
| Romania | 41.5 | 0.4 |
| Bulgaria | 28.6 | 0.9 |
| Italy | 56.1 | 0.4 |
| USA | 89.7 | 3.8 |
| Australia | 57 | 1.3 |
| South Korea | 78.8 | 1.5 |
| Colombia | 68.3 | 6.5 |
| South Africa | 78.8 | 3.5 |

Note: This data are self-report. Source: Atwoli et al., 2013; Bum et al., 2005; Gaviria et al., 2016; Kilpatrick et al., 2013; Rosenman, et al., 2002; Trautman & Witchers (2018).

## Current study

Therefore, in this study, our main aims were to measure the lifetime exposure to traumatic events in a population-based sample of Poles and to estimate the prevalence rate of probable PTSD according to DSM–5 criteria. Despite the is the lack of valid, population-based studies on PTSD prevalence based on DSM–5 criteria across the world, it should be emphasized that recent comparisons of trauma exposure and PTSD estimates in national samples of Americans using DSM–IV versus DSM–5 criteria showed that revisions in PTSD diagnosis had a minimal effect on results for the prevalence of this disorder in the general population [13]. In this study, we also analyzed the relations between sociodemographic variables and probable PTSD prevalence as well as between types of PTEs and probable PTSD prevalence. Finally, we investigated the link between PTSD intensity and levels of life satisfaction to evaluate the extent to which PTSD severity is indeed related to decreased life satisfaction–a finding reported in previous works [e.g., 30, 31], but rarely investigated in population-based studies on PTSD.

## Method

### Participants and procedure

Data were obtained by a professional company specializing in nationwide Polish research panels between September and October 2022 through a survey of a representative sample comprising 1,598 adult Poles. The study measures were sent to the participants in an online format via the company's platform. Participation in this project was anonymous and voluntary. Informed consent was obtained from all participants, who received remuneration in the form of tokens given by the survey company. The project was approved by the ethics committee of the Faculty of Psychology, University of Warsaw.

The sample consisted of participants aged 18–97 ($M$ = 48.78; $SD$ = 20.50), divided into nearly equal numbers– 810 women aged 18–96 ($M$ = 47.18; $SD$ = 19.74) and 788 men aged 18–97 ($M$ = 50.42; $SD$ = 21.14)–constituting 50.7% and 49.3% of the sample, respectively. Many

participants (40.6%) were aged 18–46. The most frequent area of residence was a village (34.7%). Many participants had higher education (44.2%). Almost half of the participants were married (49.7%). The sample of 1,598 participants allowed us to estimate the prevalence rate in the total population of 37,907,704 Poles [32] with a maximum error of 1%, assuming the sampling fraction to be equal to 0.1.

## Measures

**Trauma exposure and PTSD.** To assess exposure to potentially traumatic events and the current probable PTSD rate, we used the Polish adaptation of the Posttraumatic Diagnostic Scale for DSM–5 [PDS-5; [33]]. The PDS–5 is a self-report measure consisting of 24 items that evaluate PTSD symptom severity in the previous month according to DSM–5 criteria (see Results). This tool begins with trauma screening questions that aim to identify each participant's trauma history and to specify the traumatic event that currently most bothers the participant. After that participants answer twenty questions based on the DSM–5 symptom clusters: intrusion (Items 1–5), avoidance (Items 6–7), changes in mood and cognition (Items 8–14), and arousal and hyperreactivity (Items 15–20). The final four items are related to distress and interference caused by PTSD symptoms, as well as the beginning and duration of the symptoms. All items are presented on a 5-point scale assessing the frequency and severity of symptoms, ranging from 0 (not at all) to 4 (6 or more times a week/severe). The Polish version of the PDS–5 has very good psychometric properties [34]. According to the recommendations, we decided to use the DSM–5 diagnostic criteria instead of cut-off points because cut-off points tend to be reliable only for the countries in which they were developed [33].

In our study, to assess exposure to potentially traumatic events and current probable PTSD diagnosis based on the aforementioned questionnaire used (PDS–5) we followed the DSM-5 criteria for PDS. These criteria were operationalized as follows: Criterion A–at least one traumatic event indicated on the PDS–5 traumatic events list; Criterion B–the presence of at least one symptom indicated; Criterion C–the presence of at least one symptom indicated; Criterion D–the presence of at least two symptoms indicated; Criterion E–the presence of at least two symptoms indicated; Criterion F–the presence of indicated symptoms lasting for more than 1 month; and Criterion G–the presence of indicated symptoms disturbing daily living. Descriptive statistics of PDS-5 items in the sample are presented in S1 Appendix. Descriptive statistics of PDS-5 scores in the sample are presented in S2 Appendix.

**Subjective well-being.** Subjective well-being was evaluated using the Satisfaction with Life Scale [SWLS; [35, 36]]. The SWLS comprises five items, and the participants evaluate each item on a 7-point scale, ranging from 1 (strongly disagree) to 7 (strongly agree). Thus, a higher total score indicates a higher level of life satisfaction. The Cronbach's alpha of the Polish version of SWLS is satisfactory (.90).

**Data analysis.** First, the sample's characteristics were calculated. Next, probable PTSD diagnosis was assessed in each case, based on the DSM–5 diagnostic criteria. The prevalence of traumatic events was verified and 95% confidence intervals (CIs) based on the binomial distributions were calculated. The statistical significance of the relations between sociodemographic variables and probable PTSD prevalence was verified with the use of the Pearson chi-squared test of independence followed by a z-test based on Bonferroni correction for several comparisons. The prevalence rates of probable PTSD, depending on the type of traumatic event indicated as the most disturbing, were also calculated along with 95% CIs based on the binomial distribution. The difference between the respondents with possible PTSD diagnosis and those without a PTSD diagnosis was assessed with the use of the Student's *t*-test for independent samples and Cohen's *d* effect-size measure.

## Results

### Prevalence of potentially traumatic events

Table 2 presents the PTE prevalence rates in the current sample, with 95% CIs based on the binomial distribution.

The majority of the participants had experienced at least one traumatic event. The most prevalent type of traumatic event was a serious, life-threatening illness. The least prevalent type of traumatic event was sexual assault.

### Prevalence of PTSD

Fig 1 depicts distributions of PDS-5 scores acquired in the current sample.

The prevalence rate of probable PTSD diagnosis based on the DSM–5 criteria used in this study was 31.1% in the group of participants who indicated at least one PTE ($n = 634$), which constituted 18.8% of the total sample (95% CI [28.27, 34.12]). Table 3 depicts the prevalence rates of PTSD in the group of participants who indicated at least one traumatic event, depending on sociodemographic variables with the values of the chi-squared test for independence.

The prevalence rate of PTSD was significantly higher in the group of female participants than in the group of male participants. There were also statistically significant relations between PTSD prevalence rate and participants' ages, education levels, and relationship status. To identify the variables that differed between groups, their values were compared with adjusted $p$-values following the Bonferroni method. This method indicated that PTSD rates were significantly higher in the group of participants aged 18–46 than in the older age groups regardless of gender, significantly lower in the group of participants with higher education compared to the group with less education, and significantly lower in the group of married participants than in the group of unmarried participants.

Table 4 presents the prevalence rates of PTSD, based on the type of traumatic event indicated as the most disturbing.

The traumatic events associated with the highest rates of PTSD were child abuse and sexual assault. Accidents were associated with the lowest rate of PTSD.

### PTSD and satisfaction with life in the national sample of Poles

Satisfaction with life in the group of participants diagnosed with probable PTSD was significantly lower than in the group of participants without the diagnosis. Fig 2 depicts the mean values of

**Table 2. Prevalence rates of potentially traumatic events in the national sample of Poles (n = 1598).**

| Potentially traumatic events | n | % | 95%CI |
|---|---|---|---|
| Serious, life-threatening illness | 706 | 44.2 | 41.76–46.63 |
| Physical assault | 91 | 5.7 | 4.66–6.94 |
| Sexual assault | 83 | 5.2 | 4.21–6.40 |
| Military or combat-related | 100 | 6.3 | 5.17–7.56 |
| Child abuse | 107 | 6.7 | 5.57–8.03 |
| Accident | 304 | 19.0 | 17.17–21.02 |
| Natural disaster | 180 | 11.3 | 9.81–12.91 |
| Other | 92 | 5.8 | 4.72–7.01 |
| Total with any PTE | 964 | 60.3 | 57.90–62.70 |

*Note*: *n*–number of participants; %—current sample percentage

95% *CI*– 95% confidence interval based on the binomial distribution.

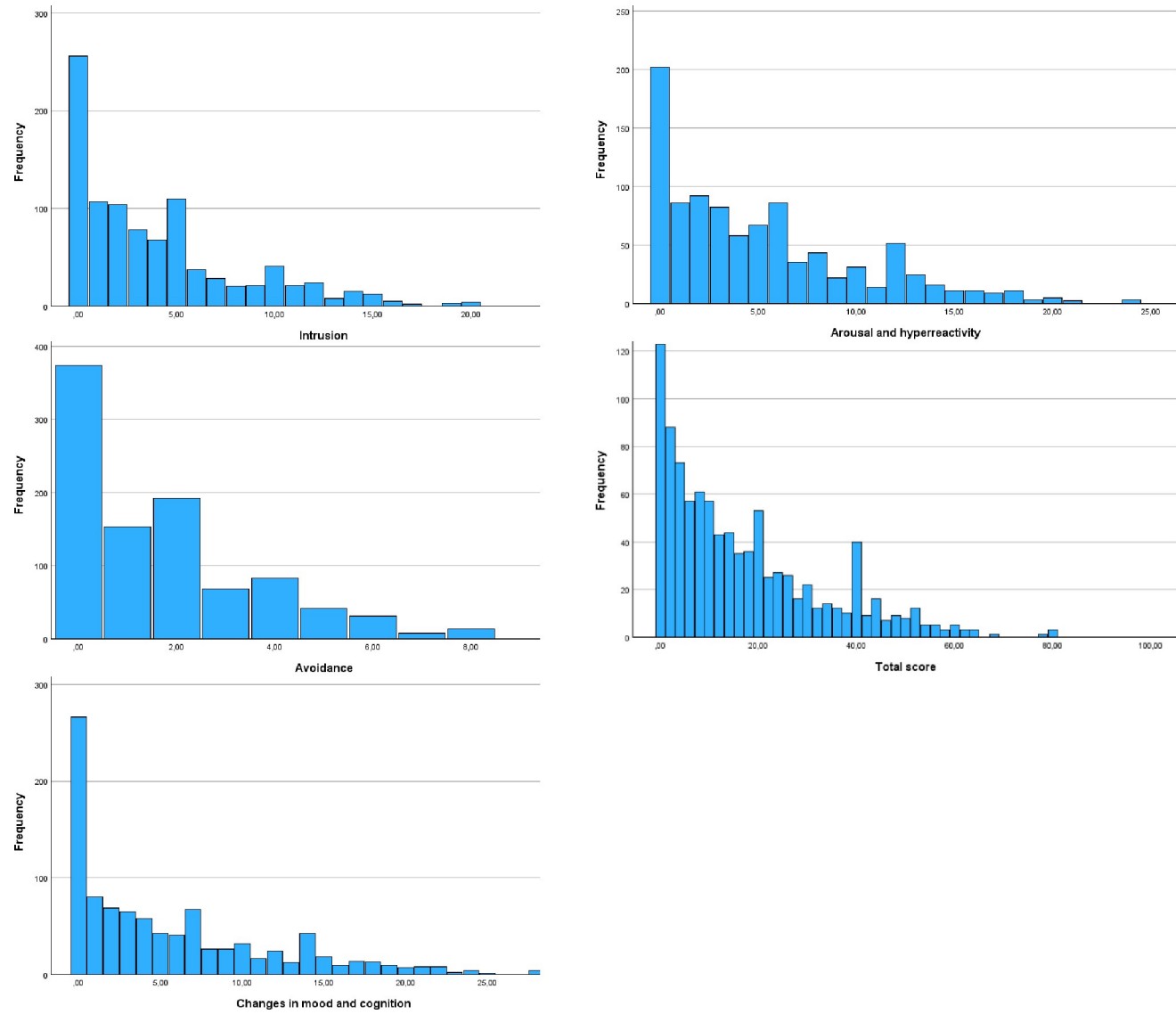

**Fig 1. Histograms for the PTSD-5 scores among the study participants.**

satisfaction with life in both groups. According to the value of the Student's *t*-test for independent samples, the difference was statistically significant, $t(414.63) = 7.67$, $p < .001$, $d = [.40, .66]$.

## Discussion

Our study assessed trauma exposure and probable post-traumatic stress disorder (PTSD) prevalence in a representative national sample of Poles. The findings showed that more than 60.3% of Poles had experienced at least one potentially traumatic event (PTE). We also found an intriguingly high level of current, probable PTSD– 31.1% in trauma survivors, who comprised 18.8% of the total study sample–suggesting that almost 20% of the Polish population is affected by probable PTSD. This data is largely in line with a recent meta-analysis published in Polish that showed a very high prevalence of PTSD, although observed only in convenience samples of Poles [29]. Our findings suggest that the current rate of probable PTSD in Poland is much

**Table 3. Prevalence rates of PTSD in the national sample of Poles depending on sociodemographic variables.**

| Sociodemographic characteristics | | PTSD prevalence | | χ2 | df | p |
|---|---|---|---|---|---|---|
| | | n | % | | | |
| Gender | Females | 187 | 37.5 | 19.49 | 1 | .001 |
| | Males | 113 | 24.3 | | | |
| Age | 18–46 | 157 | 37.1 | 12.73 | 2 | .002 |
| | 47–76 | 111 | 26.7 | | | |
| | 77+ | 32 | 25.4 | | | |
| Place of residence | a village | 96 | 30.4 | .82 | 4 | .935 |
| | a small town up to 20,000 inhabitants | 38 | 31.4 | | | |
| | a middle town 20,000 to 99,000 inhabitants | 57 | 29.8 | | | |
| | a larger city 100 to 500,000 inhabitants | 56 | 31.1 | | | |
| | a big city more than 500,000 inhabitants | 53 | 34.0 | | | |
| Education | primary | 7 | 41.2 | 10.80 | 4 | .029 |
| | vocational | 23 | 40.4 | | | |
| | secondary | 131 | 34.3 | | | |
| | incomplete higher | 19 | 37.3 | | | |
| | higher | 120 | 26.3 | | | |
| Relationship status | single | 73 | 44.0 | 20.74 | 5 | .001 |
| | married | 133 | 26.5 | | | |
| | informal relationship | 37 | 29.1 | | | |
| | separated | 3 | 25.0 | | | |
| | divorced | 14 | 27.5 | | | |
| | widow/widower | 40 | 37.7 | | | |

*Note*: *n*–number of participants; χ² –chi-squared test for independence; *df*–degrees of freedom; *p*–statistical significance.

higher in comparison to the findings reported in all of the above-mentioned studies on probable PTSD prevalence in different countries worldwide (Tables 2–4) [13, 15, 18–21]. Estimates reported in these studies were much lower than the lower bound of the CI for Poland found in this study (95% CI [28.27, 34.12]; Table 1).

These conclusions obviously must be approached with caution, as different research methods and diagnostic criteria were applied in the compared studies. Specifically, we should take

**Table 4. Prevalence rates of PTSD in the national sample of Poles depending on the type of traumatic event indicated as most disturbing.**

| Traumatic events | n | % | 95% CI |
|---|---|---|---|
| Serious, life-threatening illness | 146 | 28.1 | 24.43–32.16 |
| Physical assault | 12 | 37.5 | 22.68–55.10 |
| Sexual assault | 18 | 56.3 | 39.00–72.11 |
| Military or combat-related | 13 | 29.5 | 18.44–45.38 |
| Child abuse | 32 | 60.4 | 46.77–72.55 |
| Accident | 34 | 21.4 | 15.70–28.44 |
| Natural disaster | 19 | 28.8 | 19.18–40.79 |
| Other | 26 | 44.1 | 32.03–56.85 |

*Note*: *n*–number of participants; %—sample percentage of participants indicating the even as most disturbing; 95% *CI*– 95% confidence interval based on the binomial distribution.

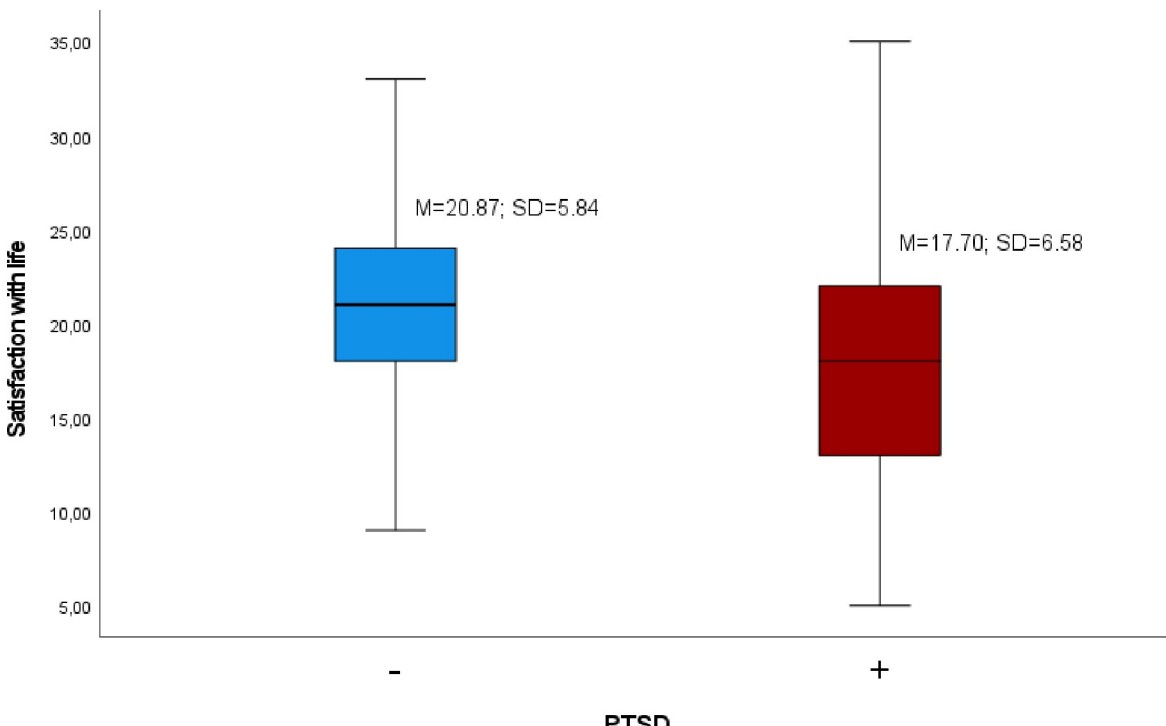

**Fig 2. Mean value of satisfaction with life in the group of participants diagnosed with PTSD compared with those without PTSD diagnosis.**

into account the methodological differences between our study, in which we followed the DSM–5 diagnostic criteria for the probable PTSD diagnosis, and the aforementioned studies, which used various, sometimes country-specific cut-off scores for the PTSD diagnosis among the study participants. However, the observed differences call for an attempt to identify possible reasons for this divergence in probable PTSD rates. First, it needs to be emphasized that apart from methodological factors, the cross-national differences in PTSD rates may also be linked to each country's socioeconomic, institutional, or historical characteristics [1, 9, 10] as well as the ongoing situation in the world. Regarding the first two factors, we should underscore the significant underfunding of the public healthcare system in Poland, resulting in limited access to mental health specialists, particularly those trained in trauma-oriented approaches [37, 38]. It is also worth focusing on the current situation related to the COVID-19 pandemic and the ongoing Russian war on Ukraine, which obviously may impact the results. Previous studies that we refer to were conducted before the COVID-19 pandemic, whereas our study was conducted after the COVID-19 outbreak. A recent international study showed that the total prevalence rate of probable PTSD in 11 European countries was much higher than before the COVID-19 pandemic [39]. Finally, it is worth mentioning that in our sample the most prevalent type of traumatic event was a serious, life-threatening illness. Several studies showed that this type of trauma can be related to very high PTSD intensity [e.g. 40–43].

However, we also believe that the impact of the tragic Polish history during and after WWII should be highlighted as a possible explanation [see also 25, 26, 44]. During WWII, Poland suffered enormous terror and lost approximately 17% of its pre-war population, which was the highest percentage among all countries involved in that war [45–47]. The socio-political situation in Poland after WWII (i.e., during the Communist regime from 1946 to 1989, the Polish population was subjected to repression) created a social environment that prevented

individuals from revealing their WWII-related traumatic experiences to other people and obtaining support and social acknowledgement [48]. In contrast, many people were treated as political enemies and experienced persecution and continued traumatisation, especially between 1946 and 1956. We think that these factors have led to substantial differences between the situation in Poland and other countries, particularly Western societies. This situation, leading to a lack of social acknowledgement of traumas, could also be responsible for the much higher prevalence of WWII-related PTSD among WWII survivors in Poland compared to other European countries. For example, while the PTSD rates among civilian survivors of WWII range from 1.9% in Austria [49] to 10.9% in Germany [50], among Polish survivors of WWII, these rates vary from 29.4% [25] to 38.3% [48] and even to 55.6% among Jewish Holocaust survivors in Poland (members of the Association of Children of the Holocaust) [44]. On the other hand, there is a massive body of literature on the issue of intergenerational transmission of trauma [for reviews and meta-analyses, see 51–53]; this has been empirically investigated in Poland as well, although only in convenience samples [48, 54, 55]. In particular, the problem of the potential transmission of this historic trauma through generations of Polish families may be exacerbated by the lack of social acknowledgement and linked with this, states of helplessness and hopelessness on both individual and collective levels (Figs 1, 2) [48].

It is worth emphasizing that the most recent models of PTSD have shifted from the traditionally studied individual risk factors [5, 56] to more social and interpersonal phenomena [57]. Specifically, Maercker and Horn [57] created the socio-interpersonal model of PTSD, which centers on the social acknowledgement of trauma, defined as the extent to which a trauma survivor perceives and feels social empathy and understanding of their traumatic experiences expressed by intimate partners, families, local communities, or even the whole society. Polish survivors of WWII were unable to share their war-related trauma memories in a stable, safe, and supportive community, which is a key factor in PTSD treatment and prevention [57]. They were, therefore, unable to mourn their loss, which is one of the major elements of working through a traumatic experience [58].

Regarding the structure of the findings on trauma exposure and probable PTSD prevalence, we found that the PTSD rate was significantly higher in the group of female participants than in their male counterparts. This result is in line with those of numerous studies on gender differences in PTSD, which showed that despite men's greater exposure to traumatic events, women were at a much higher risk of PTSD [for a review, see 59]. The above-mentioned trend is related to a variety of factors, including particular types of PTEs being more often experienced by women/girls (in particular, sexual trauma), stronger perceptions of threat when experiencing trauma compared with men, or gender differences in psychobiological reactions to trauma. We also found that younger participants declared higher PTSD levels than older respondents. The studies on the effect of age on PTSD prevalence have mixed results but usually present no significant effect of age on PTSD in population-based studies [e.g., 60, 61]. However, Norris et al. [62] showed that the effect of age on PTSD may vary based on culture. Furthermore, in our sample, PTSD was significantly lower in the group of participants with higher education than in the group with less education and in the group of married participants than in the group of singles, which illustrates that these sociodemographic factors may act as PTSD buffers [for meta-analyses, see, e.g., 4, 5, 7]. Lastly, while in our sample the most prevalent type of PTE was a serious, life-threatening illness and the least common was sexual assault, this latter PTE (along with child abuse) was related to the highest rates of PTSD. Sexual trauma experienced not only in adulthood but also in previous stages of life, along with other forms of child abuse, emerged as the most toxic PTE–that is, related to the highest risk of PTSD [for meta-analyses, see 12, 63]. This result may also be related to the lack of social acknowledgement and proper care for survivors [38].

Finally, consistent with our expectations, the level of life satisfaction in the group of participants diagnosed with probable PTSD was significantly lower than in the group of participants without the diagnosis. Numerous authors have observed that PTSD symptoms significantly diminish people's psychological well-being, including their subjective assessments of their lives [e.g., 30, 64]. People with PTSD demonstrate high levels of dissatisfaction across multiple life domains, including social and occupational functioning and physical health [65], as well as mental health and well-being [31]. Some authors have even posited that such strong negative associations between these two terms suggest that PTSD and life satisfaction should be regarded as two separate and opposite theoretical constructs [66, 67].

## Strengths and limitations

This study has several strengths, the primary one being that it is the first attempt to examine probable PTSD prevalence according to the DSM–5 criteria, using a world-renowned measure (PDS-5), in a nationally representative sample of Poles. However, a few limitations should be underlined as well. First of all, we used a self-report measurement of trauma exposure and PTSD, so we did not include a clinician-administered evaluation of PTSD, which is usually treated as the gold standard of PTSD assessment. Nevertheless, this kind of PTSD measurement would be very difficult to implement in this type of epidemiological study. Also, the cross-sectional nature of this study precludes the drawing of causal conclusions on the link between PTSD and life satisfaction.

Furthermore, different research methods and diagnostic criteria for the probable PTSD diagnosis were applied in our particular research compared to other studies mentioned in this article, which may be the reason for different PTSD prevalence rates across countries. Specifically, in our study belonging to the group of people with a probable diagnosis of PTSD was determined based on DSM-5 diagnostic criteria, which corresponds to an individual configuration of items in the PDS-5 questionnaire. However, when including the subjects in the group of people with probable PTSD diagnosis, we took into account not only the occurrence of symptoms but also their duration (criterion F) and—what is important—the impact on the functioning of the subjects (criterion G). By doing this, we may be able to "protect" our results from overestimating the level of probable PTSD diagnosis in our sample. Finally, this study was also conducted against the backdrop of a specific global context (the COVID-19 pandemic and, especially, the Russian war on Ukraine), which may affect the results.

## Conclusion

Despite these limitations, our study makes an important contribution to research on trauma and probable PTSD, in both the Polish and global contexts. Specifically, we confirmed that the current rate of probable PTSD in Poland is intriguingly high in comparison to rates in other countries. Our research may motivate further studies investigating cross-national differences in PTSD ratings and trauma exposure, with an emphasis on the analysis of each country's socioeconomic, institutional, and historical characteristics. In our opinion, exploring PTSD predictors from such a vast interdisciplinary perspective could provide further insights into the nature of PTSD and trauma resilience in a cross-cultural context [8].

## Supporting information

**S1 Appendix. Descriptive statistics of PDS-5 items (raw scores and scores dichotomized into symptoms).**
(TIF)

**S2 Appendix. Descriptive statistics of PDS-5 scores.**
(TIF)

**S1 Data.**
(XLSX)

## Author Contributions

**Conceptualization:** Marcin Rzeszutek, Małgorzata Dragan, Maja Lis-Turlejska, Katarzyna Schier, Paweł Holas, Katarzyna Drabarek, Angelika Van Hoy, Małgorzata Pięta, Cecylia Poncyliusz, Magdalena Michałowska, Gabriela Wdowczyk, Natalia Borowska.

**Data curation:** Marcin Rzeszutek, Małgorzata Dragan, Szymon Szumiał.

**Formal analysis:** Maja Lis-Turlejska, Katarzyna Drabarek, Szymon Szumiał.

**Funding acquisition:** Marcin Rzeszutek, Małgorzata Dragan.

**Investigation:** Katarzyna Drabarek, Małgorzata Pięta.

**Methodology:** Marcin Rzeszutek, Małgorzata Dragan, Maja Lis-Turlejska, Katarzyna Schier, Angelika Van Hoy.

**Resources:** Marcin Rzeszutek.

**Supervision:** Marcin Rzeszutek, Paweł Holas.

**Writing – original draft:** Marcin Rzeszutek, Małgorzata Dragan, Maja Lis-Turlejska, Katarzyna Schier, Paweł Holas, Katarzyna Drabarek, Angelika Van Hoy, Małgorzata Pięta, Cecylia Poncyliusz, Magdalena Michałowska, Gabriela Wdowczyk.

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
