## [Decision Letter · Decision Letter 0]

17 Mar 2023

PONE-D-23-05085Exposure to Potentially Traumatic Events and Probable PTSD in a National Sample of Poles:

Why Does Poland’s PTSD Prevalence Differ from Other National Estimates?PLOS ONE

Dear Dr. Rzeszutek,

Thank you for submitting your manuscript to PLOS ONE. After careful consideration, we feel that it has merit but does not fully meet PLOS ONE’s publication criteria as it currently stands. Therefore, we invite you to submit a revised version of the manuscript that addresses the points raised during the review process. Both reviewers have expressed interest in your manuscript, and I share their enthusiasm for your contribution to the literature. However, Reviewer 2 raised some serious concerns regarding the scoring procedure for the probable PTSD diagnosis, which does not adhere to the DSM-5 criteria. Therefore, before accepting your work, I kindly request that you address each of the points raised by the reviewers and revise the concerning parts of your manuscript accordingly.

We look forward to receiving your revised manuscript.

Kind regards,

Inga Schalinski

Academic Editor

PLOS ONE

Journal Requirements:

2. Please provide additional details regarding ethical approval in the body of your manuscript. In the Methods section, please ensure that you have specified the name of the IRB/ethics committee that approved your study

3. We note that you have stated that you will provide repository information for your data at acceptance. Should your manuscript be accepted for publication, we will hold it until you provide the relevant accession numbers or DOIs necessary to access your data. If you wish to make changes to your Data Availability statement, please describe these changes in your cover letter and we will update your Data Availability statement to reflect the information you provide

4. "In your Data Availability statement, you have not specified where the minimal data set underlying the results described in your manuscript can be found. PLOS defines a study's minimal data set as the underlying data used to reach the conclusions drawn in the manuscript and any additional data required to replicate the reported study findings in their entirety. All PLOS journals require that the minimal data set be made fully available. For more information about our data policy, please see http://journals.plos.org/plosone/s/data-availability.

Additional Editor Comments (if provided):

The authors need to correct the applied algorithm, because for PTSD two or more symptoms are required for each criteria: D and E. This is a major concern that may affect the reported prevalence of PTSD, title, and discussion.

Please provide more information about the trauma screening of the PDS-5: How many different types of events were assessed, is there any information of the type of confrontation? How did you ensure that “others” are in line with the definition of traumatic events based on the DSM-5.

The paragraph “PTSD Diagnostic Criteria” appears twice in the manuscript and should be removed from the results section.

Reviewers' comments:

Reviewer's Responses to Questions

**Comments to the Author**

1. Is the manuscript technically sound, and do the data support the conclusions?

Reviewer #1: Yes

Reviewer #2: Partly

2. Has the statistical analysis been performed appropriately and rigorously? 

Reviewer #1: Yes

Reviewer #2: Yes

3. Have the authors made all data underlying the findings in their manuscript fully available?

Reviewer #1: Yes

Reviewer #2: No

4. Is the manuscript presented in an intelligible fashion and written in standard English?

Reviewer #1: Yes

Reviewer #2: Yes

5. Review Comments to the Author

Reviewer #1: The paper is Interesting and important in general, not only for Polish readers. It describes the representative prevalence of PTSD and the satisfaction in life which are significant data for any comparative study.

I suggest only minor changes in the structure and other in discussion, however depending to the Authors decision.

I suggest to remove the repetition of PTSD diagnostic criteria and put it only in one place.

Authors discuss the high result of the PTSD prevalence in the examined group. In my opinion the easiest explanation is that Authors used DSM-5 criteria but compared with the publications based on more restrictive DSM-IVTR or DSM-IIIR editions.

Discussion is limited to self-quotation but doesn’t mention other researchers as i.e. most known Polish-Jewish Holocaust survivors researcher Maria Orwid.

In general the correlation between PTSD and less satisfaction in life in groups confirm the results.

Reviewer #2: The authors use an established self-report instrument to assess trauma burden, PTSD symptoms and quality of life in a sample representative of the Polish population. The following are the core statements of the article:

Both the number of people reporting at least one trauma and the number of self-reported PTSD sufferers are comparatively high

The most frequently reported trauma type is experiencing a severe physical illness

The trauma types with the highest likelihood of PTSD are sexual abuse, sexual violence and physical violence

In the PTSD-group young age (18 - 46 years), female gender, low educational status and being single are more prevalent than in the no-PTSD- group

In comparison, the quality of life is significantly lower in the PTSD-group

The study makes an important contribution to identifying the prevalence of PTSD and its specifics in Poland. The representative sample for the Polish population should be emphasized here. In addition, the theoretical and empirical introduction to the research question is very successful and pleasant to read.

There are three major aspects that need to be addressed:

1)The argumentation not to work with the American cut-off of the PDS -5 value is meaningful. At the same time, I find the omission of a quantitative evaluation problematic. A quantitative evaluation would provide us with important information to better interpret the high symptoms recorded in the self-report. The authors do not report how high the scale score must be in order to fulfil one symptom per PTSD criterion. To my knowledge, the PDS -5 counts each item >0. Thus, according to the authors' approach, individuals with a total scale score ≥ 7 would be part of the PTSD-group (comparison: US cut-off score =28). This is difficult against the background of high overlaps of many PTSD symptoms with symptoms of other mental illnesses as well as the fact that it could represent non-clinically relevant manifestations of symptoms. This approach carries the risk of overestimating the prevalence of PTSD. In addition to the evaluation already implemented, a possible approach is to carry out a quantitative evaluation and to present this graphically by means of violin charts or box plots for the total value as well as all subscales.

2)With regard to the sample, the following points would be important:

a. For which characteristics is the sample representative?

b. What is their recruitment process? How many of the original people requested did not participate in the study?

3)In the discussion, relevant aspects are well addressed. In my view, one important aspect should be given more attention. How can the excessively high rate of serious physical illness as trauma be explained and the large proportion of the PTSD group.

Furthermore, I see the following ways to improve the manuscript:

In the title you use the word potentially. Here I would suggest speaking of the self-report

Please describe your data preparation: Have there been missing values? Have there been outliers? How have you delt with them?

Please describe the psychometric quality of the SWLS and name studies on this subject.

Please name the use of the Bonferroni correction already in the methods section.

Table 1:

Here you name the prevalences in different countries. How did you arrive at the percentages? As you state in the theory section, there are usually several studies with different prevalences. Either you give the range and/or the median in the case of very different values or the mean in the case of similar values.

I would recommend including in the notes of the table that it is self-report data.

Chart 1:

Please start the Y-axis at zero. The differences between the groups will otherwise be overrepresented.

Again, I would recommend using a graph to map the distribution. Violin charts or boxplots would again be suitable. Alternatively, you could put this in the supplement for the interested reader. In you decide not to integrate it, please plot the standard deviation lines in the chart.

Among other things, you repeat the operationalisation of the PTSD diagnosis in the results section. I would suggest avoiding repetitions and only include it in the methods section

There is a typo on page 4

6. PLOS authors have the option to publish the peer review history of their article (what does this mean?). If published, this will include your full peer review and any attached files.

Reviewer #1: **Yes: **Krzysztof Rutkowski

Reviewer #2: **Yes: **Sarah Wyka

---

## [Author Response · Author response to Decision Letter 0]

6 Apr 2023

Dear Editor, Dear Reviewers, 

Thank you very much for your suggestions and remarks concerning our article titled “Exposure to Self-Report Traumatic Events and Probable PTSD in a National Sample of Poles:

Why Does Poland’s PTSD Prevalence Differ from Other National Estimates?”, which we would like to publish in PLOS ONE. We referred to all reviewers’ remarks. Below we cite every statement and comment of the reviewers and provide their answers in parentheses. 

Editor’s remarks

[Thank you for this remark. I can assure that the whole manuscript is in line with PLOS ONE style.]

2. Please provide additional details regarding ethical approval in the body of your manuscript. In the Methods section, please ensure that you have specified the name of the IRB/ethics committee that approved your study.

[We have included this additional information in the Method section.]

3. We note that you have stated that you will provide repository information for your data at acceptance. Should your manuscript be accepted for publication, we will hold it until you provide the relevant accession numbers or DOIs necessary to access your data. If you wish to make changes to your Data Availability statement, please describe these changes in your cover letter and we will update your Data Availability statement to reflect the information you provide

[Thank you for this information. We have included relevant change of this statement in the cover letter.]

4. "In your Data Availability statement, you have not specified where the minimal data set underlying the results described in your manuscript can be found. PLOS defines a study's minimal data set as the underlying data used to reach the conclusions drawn in the manuscript and any additional data required to replicate the reported study findings in their entirety. All PLOS journals require that the minimal data set be made fully available. For more information about our data policy, please see http://journals.plos.org/plosone/s/data-availability.

[Thank you for this remark. We included relevant information about this in the cover letter.]

The authors need to correct the applied algorithm, because for PTSD two or more symptoms are required for each criteria: D and E. This is a major concern that may affect the reported prevalence of PTSD, title, and discussion.

[Thank you very much for this remark. We are very grateful for this, as we found that it was some small typo/mistake in the description of these criteria in the Results section. Now in the revised version of the manuscript we modified the results section. This mistake was only present in the description of the results, and not the applied algorithm of the data, which is calculated correctly. In order to be sure, we doubled-checked this issue, and we can confirm that there is no error in the algorithm.]

Please provide more information about the trauma screening of the PDS-5: How many different types of events were assessed, is there any information of the type of confrontation? How did you ensure that “others” are in line with the definition of traumatic events based on the DSM-5.

[Thank you very much for this remark. The PDS-5 is a 24-item self-report measure that assesses PTSD symptom severity in the last month according to DSM-5 criteria. An item for each of the 20 DSM-5 PTSD symptoms is included, and an additional four items asking about distress and interference caused by PTSD symptoms as well as onset and duration of symptoms.

Symptom items are rated on a 5-point scale of frequency and severity ranging from 0 (Not at all) to 4 (symptom present 6 or more times a week / severe). The types of traumatic events are as following (participants did not include “other” in our sample):

- Serious, life-threatening illness (e.g. heart attack)

- Assault (armed robbery, serious injury in a fight, someone pointed a gun at you, etc.)

- Sexual assault (rape, attempted rape, forced intercourse with e.g. weapons)

- Participating in hostilities or staying in an area of such hostilities

- Childhood abuse (beating, sexual acts with someone 5 or more years older)

- Accident (serious injury or death in a car accident, fire or other domestic accident, etc.)

- Natural disaster (strong hurricane, flood, earthquake, etc.)

- Other traumatic experience (please briefly describe).

The paragraph “PTSD Diagnostic Criteria” appears twice in the manuscript and should be removed from the results section.

[Thank you for this remark. We removed double descriptions of the “PTSD Diagnostic Criteria”.]

Reviewer #1: The paper is Interesting and important in general, not only for Polish readers. It describes the representative prevalence of PTSD and the satisfaction in life which are significant data for any comparative study. I suggest only minor changes in the structure and other in discussion, however depending to the Authors decision.

[Thank you very much for the nice words on our submission.]

I suggest to remove the repetition of PTSD diagnostic criteria and put it only in one place.

[It is very important remark – we removed the repetition of PTSD diagnostic criteria.]

Authors discuss the high result of the PTSD prevalence in the examined group. In my opinion the easiest explanation is that Authors used DSM-5 criteria but compared with the publications based on more restrictive DSM-IVTR or DSM-IIIR editions.

[Thank you very much for this remark. However, we cannot agree entirely with this remark, as large epidemiological study conducted by Kilpatrick et al. (2013) showed that the PTSD prevalence estimates in light of DSM-5 are even lower than their PTSD estimates in light of DSM-IV counterparts. 

Kilpatrick, D. G., Resnick, H. S., Milanak, M. E., Miller, M. W., Keyes, K. M., & Friedman, M. J. (2013). National estimates of exposure to traumatic events and PTSD prevalence using DSM-IV and DSM-5 criteria. Journal of Traumatic Stress, 26(5), 537–547. https://doi.org/10.1002/jts.21848

So it is not rather the issue of stringent vs. non-stringent PTSD levels in the subsequent DSM editions, but the case of extraordinary high level of PTSD in Poland, which was found also in the previous studies, yet not conducted in the representative sample as in our current study. ]

Dragan, M., Lis-Turlejska, M., Popiel, A., Szumiał, S., & Dragan, W. (2012). The validation of the Polish version of the Posttraumatic Diagnostic Scale and its factor structure. European Journal of Psychotraumatology, 3, Article 18479. https://doi.org/10.3402/ejpt.v3i0.18479

Popiel, A., Zawadzki, B., Pragłowska, E., & Teichman, Y. (2015). Prolonged exposure, paroxetine and the combination in the treatment of PTSD following a motor vehicle accident. A randomized clinical trial – The “TRAKT” study. Journal of Behavior Therapy and Experimental Psychiatry, 48, 17–26. https://doi.org/10.1016/j.jbtep.2015.01.002

Discussion is limited to self-quotation but doesn’t mention other researchers as i.e. most known Polish-Jewish Holocaust survivors researcher Maria Orwid.

[Thank you very much for this remark. We cited the aforementioned researcher in the revised manuscript.]

In general the correlation between PTSD and less satisfaction in life in groups confirm the results.

[Thank you for this remark.]

Reviewer #2: The authors use an established self-report instrument to assess trauma burden, PTSD symptoms and quality of life in a sample representative of the Polish population. The following are the core statements of the article: Both the number of people reporting at least one trauma and the number of self-reported PTSD sufferers are comparatively high The most frequently reported trauma type is experiencing a severe physical illness. The trauma types with the highest likelihood of PTSD are sexual abuse, sexual violence and physical violence. In the PTSD-group young age (18 - 46 years), female gender, low educational status and being single are more prevalent than in the no-PTSD- group. In comparison, the quality of life is significantly lower in the PTSD-group

The study makes an important contribution to identifying the prevalence of PTSD and its specifics in Poland. The representative sample for the Polish population should be emphasized here. In addition, the theoretical and empirical introduction to the research question is very successful and pleasant to read.

[Thank you very much for the kind words on our submission.]

There are three major aspects that need to be addressed:

1)The argumentation not to work with the American cut-off of the PDS -5 value is meaningful. At the same time, I find the omission of a quantitative evaluation problematic. A quantitative evaluation would provide us with important information to better interpret the high symptoms recorded in the self-report. The authors do not report how high the scale score must be in order to fulfil one symptom per PTSD criterion. To my knowledge, the PDS -5 counts each item >0. Thus, according to the authors' approach, individuals with a total scale score ≥ 7 would be part of the PTSD-group (comparison: US cut-off score =28). This is difficult against the background of high overlaps of many PTSD symptoms with symptoms of other mental illnesses as well as the fact that it could represent non-clinically relevant manifestations of symptoms. This approach carries the risk of overestimating the prevalence of PTSD. In addition to the evaluation already implemented, a possible approach is to carry out a quantitative evaluation and to present this graphically by means of violin charts or box plots for the total value as well as all subscales.

[Thank you very much for this remark. However, we feel that there is some misunderstanding. In our study we performed quantitative evaluation of PTSD in light of PDS-5, but not via the cut-off score, but according to DSM-5 criteria. The PDS-5 is a 24-item self-report measure that assesses PTSD symptom severity in the last month according to DSM-5 criteria. An item for each of the 20 DSM-5 PTSD symptoms is included, and an additional four items ask about distress and interference caused by PTSD symptoms as well as onset and duration of symptoms.

Symptom items are rated on a 5-point scale of frequency and severity ranging from 0 (Not at all) to 4 (symptoms present 6 or more times a week / severe). 

In order to clarify the issue of quantitative evaluation we included a table with Posttraumatic Diagnostic Scale for DSM-5 items’ means and SDs for our sample in the Appendix A.

We think that the remark of the Reviewer is extremely important in terms of the interpretation of the results. In the future research it would be valuable to look for the overlap of many PTSD symptoms with symptoms of other mental illnesses in the Polish sample.]

2)With regard to the sample, the following points would be important:

a. For which characteristics is the sample representative?

b. What is their recruitment process? How many of the original people requested did not participate in the study?

[Thank you very much for this remark. The sample was representative reflecting the characteristics of the Polish population (e.g. gender, age, place of residence, education) according to the data gathered by 

Main Statistical Office in Poland. (2020) https://demografia.stat.gov.pl/BazaDemografia/StartIntro.aspx , and thus followed the standards of conducting representative studies. 

In particular, the sample of 1,598 participants allowed us to estimate the prevalence rate in the total population of 37,907,704 Poles (see Main Statistical Office 2020) with a maximum error of 1%, assuming the sampling fraction to be equal to 0.1.

Finally, the external company, which was responsible for conducting the study and specializes in the nationwide Polish research panels sent the invitation to about 8000 Poles so as to reach the final, representative sample in our particular study of 1598 Poles.]

3) In the discussion, relevant aspects are well addressed. In my view, one important aspect should be given more attention. How can the excessively high rate of serious physical illness as trauma be explained and the large proportion of the PTSD group.

[Thank you very much for this remark. It is difficult to directly indicate if the high rate of serious physical illness as trauma can be linked to excessive high PTSD in our sample, however we included appropriate citations on that problem in the manuscript.]

Furthermore, I see the following ways to improve the manuscript:

In the title you use the word potentially. Here I would suggest speaking of the self-report

[Thank you very much for this remark. We modified the title according to this suggestion.]

Please describe your data preparation: Have there been missing values? Have there been outliers? How have you delt with them?

[Thank you very much for this remark. In the revised version of the Method section we described in more detail the data gathering by the external company. The sample was representative according to the Main Statistical Office in Poland. (2020), including the standards of conducting representative studies (e.g. gender, age, place of residence, education) https://demografia.stat.gov.pl/BazaDemografia/StartIntro.aspx

In particular, the sample of 1,598 participants allowed us to estimate the prevalence rate in the total population of 37,907,704 Poles (see Main Statistical Office 2020) with a maximum error of 1%, assuming the sampling fraction to be equal to 0.1.

Finally, the external company, which was responsible for conducting the study sent the invitation to about 8000 Poles so as to reach the final, representative sample in our particular study of 1598 Poles.]

Please describe the psychometric quality of the SWLS and name studies on this subject.

[Thank you for this remark. We included this information.]

Please name the use of the Bonferroni correction already in the methods section.

[We added the comment on the use of Bonferroni correction to the Data Analysis section.]

Table 1:

Here you name the prevalences in different countries. How did you arrive at the percentages? As you state in the theory section, there are usually several studies with different prevalences. Either you give the range and/or the median in the case of very different values or the mean in the case of similar values. I would recommend including in the notes of the table that it is self-report data.

[Thank you for this remark. This data was taken from the below references, which we mentioned in the references lists. We also included in the notes of the table that it is self-reported data.

Source: Atwoli et al., 2013; Bum et al., 2005; Gaviria et al., 2016; Kilpatrick et al., 2013; Rosenman, et al., 2002; Trautman & Witchers (2018).

Chart 1:

Please start the Y-axis at zero. The differences between the groups will otherwise be overrepresented.

Again, I would recommend using a graph to map the distribution. Violin charts or boxplots would again be suitable. Alternatively, you could put this in the supplement for the interested reader. In you decide not to integrate it, please plot the standard deviation lines in the chart.

[We replaced the former figure with boxplots as recommended.]

Among other things, you repeat the operationalisation of the PTSD diagnosis in the results section. I would suggest avoiding repetitions and only include it in the methods section

[Yes, it was a typo/mistake – we corrected it.]

There is a typo on page 4

[Thank you, we corrected this typo.]

To sum up, I would like to thank the Editor and Reviewers again for their time and effort. I found all the comments very useful, and I believe that they helped me to improve the manuscript's quality. I sincerely appreciate the chance you gave me to revise and submit it to be considered for publication in PLOS ONE.

---

## [Decision Letter · Decision Letter 1]

28 Apr 2023

PONE-D-23-05085R1Exposure to Self-Report Traumatic Events and Probable PTSD in a National Sample of Poles:

Why Does Poland’s PTSD Prevalence Differ from Other National Estimates?PLOS ONE

Dear Dr. Rzeszutek,

Thank you for submitting your manuscript to PLOS ONE. After careful consideration, we feel that it has merit but does not fully meet PLOS ONE’s publication criteria as it currently stands. Therefore, we invite you to submit a revised version of the manuscript that addresses the points raised during the review process.

We look forward to receiving your revised manuscript.

Kind regards,

Inga Schalinski

Academic Editor

PLOS ONE

Additional Editor Comments:

I have noticed that some of the responses were not sufficient to accept the manuscript in its current form. I have highlighted the points that are necessary to accept your manuscript.

In response to Reviewer 2's suggestion, please include a visual representation of the distribution of PDS total scores (e.g., histogram, bean plot) in the results section. Furthermore, you should discuss the scoring criteria in light of cut-off values employed in other studies, thereby acknowledging methodological differences and enabling readers to better understand your findings.

Specifically, the fact that a total score of ≥ 7 is sufficient to assign individuals to the group of probable PTSD in the current study (using DSM-5 criteria) in contrast to cut-offs of ≥ 28 (in addition to the DSM-5 criteria) used in other samples and countries. This comparison could be made explicit in the discussion sections of the paper, highlighting the methodological difference and potential impact on the comparison of prevalence rates across countries. This could help readers interpret the findings of the current study and understand the potential impact of the scoring criteria on prevalence rates reported across studies.

Overall, addressing Reviewer 2's recommendation could help strengthen the methodological rigor of the study and improve the interpretation and generalizability of the findings.

Please refer to probable PTSD throughout the manuscript.

Minors:

Please provide the information about the trauma screening in your manuscript.

Please shift the operationalisation for potential PTSD to the methods sections instead of the result section.

Reviewers' comments:

Reviewer's Responses to Questions

**Comments to the Author**

1. If the authors have adequately addressed your comments raised in a previous round of review and you feel that this manuscript is now acceptable for publication, you may indicate that here to bypass the “Comments to the Author” section, enter your conflict of interest statement in the “Confidential to Editor” section, and submit your "Accept" recommendation.

Reviewer #1: All comments have been addressed

Reviewer #2: (No Response)

2. Is the manuscript technically sound, and do the data support the conclusions?

Reviewer #1: Yes

Reviewer #2: Partly

3. Has the statistical analysis been performed appropriately and rigorously? 

Reviewer #1: Yes

Reviewer #2: Yes

4. Have the authors made all data underlying the findings in their manuscript fully available?

Reviewer #1: Yes

Reviewer #2: Yes

5. Is the manuscript presented in an intelligible fashion and written in standard English?

Reviewer #1: Yes

Reviewer #2: Yes

6. Review Comments to the Author

Reviewer #1: I have no comments to the Authors. The paper is interesting and worth of publication. All previous comments have been addressed.

Reviewer #2: The authors have incorporated key points, such as outlining the representativeness of the sample. The main research question remains highly relevant and the results make an important contribution to PTSD research.

However, there is a critical point that they have not adequately addressed, namely the use of a cut-off of ∑=6 and its potential content risks, which may lead to an overestimation of PTSD prevalence. While it is reasonable not to use the US cut-off, the use of such a low cut-off is not conclusive. It should also be noted that a score of 1 on the Likert scale of the PDS-5 means "once a week or less/a little," which means that a person might meet criterion B for example if they think about the incident once a month. As a result, subsyndromal manifestations of the criteria and disorder may not be adequately screened out. Especially since there is considerable overlap between PTSD symptoms and those of other disorders, such as depression. Again, a cut-off that is too low results in a significant overestimation of PTSD prevalence.

The authors have provided a table of descriptive statistics at the item level, but the point of criticism raised has not been sufficiently addressed. It would be helpful to include mean values, standard deviations, or median and quartiles, as well as extreme values, depending on the distribution, at both the criterion level and the total score level. This would enable readers to see how many people in the sample have very low values, which could be important for understanding the prevalence of subsyndromal PTSD. In addition, a visual representation by means of a scatter diagram would be even more helpful.

7. PLOS authors have the option to publish the peer review history of their article (what does this mean?). If published, this will include your full peer review and any attached files.

Reviewer #1: **Yes: **prof. Krzysztof Rutkowski

Reviewer #2: **Yes: **Sarah Wyka

---

## [Author Response · Author response to Decision Letter 1]

10 May 2023

Dear Editor, Dear Reviewers, 

Thank you very much for your suggestions and remarks concerning our article titled “Exposure to Self-Report Traumatic Events and Probable PTSD in a National Sample of Poles: Why Does Poland’s PTSD Prevalence Differ from Other National Estimates?”, which we would like to publish in PLOS ONE. We referred to all reviewers’ remarks. Below we cite every reviewer's statement and comment and provide their answers in parentheses. All the revised parts are highlighted in red font in the revised manuscript.

Editor’s remarks

• A marked-up copy of your manuscript that highlights changes to the original version. You should upload this as a separate file labeled 'Revised Manuscript with Track Changes'.

[Thank you very much for reminding us about the formal requirements in Plos One. We have included all the aforementioned parts in the second revision of our manuscript.]

I have noticed that some of the responses were not sufficient to accept the manuscript in its current form. I have highlighted the points that are necessary to accept your manuscript. In response to Reviewer 2's suggestion, please include a visual representation of the distribution of PDS total scores (e.g., histogram, bean plot) in the results section. Furthermore, you should discuss the scoring criteria in light of cut-off values employed in other studies, thereby acknowledging methodological differences and enabling readers to better understand your findings. Specifically, the fact that a total score of ≥ 7 is sufficient to assign individuals to the group of probable PTSD in the current study (using DSM-5 criteria) in contrast to cut-offs of ≥ 28 (in addition to the DSM-5 criteria) used in other samples and countries. This comparison could be made explicit in the discussion sections of the paper, highlighting the methodological difference and potential impact on the comparison of prevalence rates across countries. This could help readers interpret the findings of the current study and understand the potential impact of the scoring criteria on prevalence rates reported across studies. Addressing Reviewer 2's recommendation could help strengthen the methodological rigor of the study and improve the interpretation and generalizability of the findings.

[Thank you very much for summarizing the Reviewer's 2 remarks. As you may see below, we have introduced all his/her remarks accordingly. However, we would like to draw your attention to one point again. In our study we performed a quantitative evaluation of PTSD in light of PDS-5, but not via the cut-off score, but according to DSM-5 criteria. According to the recommendations (see Foa et al., 2016), we decided to use the DSM–5 diagnostic criteria instead of cut-off points because cut-off points tend to be reliable only in the countries in which they were developed. In other words, in the analyzes presented in this article, belonging to the group of people with a probable diagnosis of PTSD was determined on the basis of DSM-5 diagnostic criteria. These criteria correspond to individual items of the PDS-5 questionnaire. We understand the reviewer's remark on this issue and have reflected this in the discussion and limitations section.]

Please refer to probable PTSD throughout the manuscript.

[We used this phrase throughout the manuscript.]

Minors:

Please provide the information about the trauma screening in your manuscript.

[We provided this information in the Method section, i.e. in the Measures section. The trauma screening was based on the procedure described in the PDS-5 tool used in this study.]

Please shift the operationalisation for potential PTSD to the methods sections instead of the result section.

[We moved this information to the methods section.]

Reviewer #1: I have no comments to the Authors. The paper is interesting and worth of publication. All previous comments have been addressed.

[Thank you very much for your nice words about our revision.]

Reviewer #2: The authors have incorporated key points, such as outlining the representativeness of the sample. The main research question remains highly relevant and the results make an important contribution to PTSD research.

[Thank you very much for the kind words about our submission.]

However, there is a critical point that they have not adequately addressed, namely the use of a cut-off of ∑=6 and its potential content risks, which may lead to an overestimation of PTSD prevalence. While it is reasonable not to use the US cut-off, the use of such a low cut-off is not conclusive. It should also be noted that a score of 1 on the Likert scale of the PDS-5 means "once a week or less/a little," which means that a person might meet criterion B for example if they think about the incident once a month. As a result, subsyndromal manifestations of the criteria and disorder may not be adequately screened out. Especially since there is considerable overlap between PTSD symptoms and those of other disorders, such as depression. Again, a cut-off that is too low results in a significant overestimation of PTSD prevalence.

[Thank you very much for these remarks. However, we would like to emphasize one point again. In our study we performed a quantitative evaluation of PTSD in light of PDS-5, but not via the cut-off score, but according to DSM-5 criteria. According to the recommendations (see Foa et al., 2016), we decided to use the DSM–5 diagnostic criteria instead of cut-off points because cut-off points tend to be reliable only in the countries in which they were developed. In other words, in the analyzes presented in this article, belonging to the group of people with a probable diagnosis of PTSD was determined on the basis of DSM-5 diagnostic criteria. These criteria correspond to individual items of the PDS-5 questionnaire. We understand the reviewer's remark on this issue and have reflected this in the discussion and limitations section.

More specifically, the term "cut-off score" is usually understood as the threshold value of the sum of points in the questionnaire from which (>=) or above which (>) the subjects are included in the analyzed category, in this case the group of people with a probable diagnosis of PTSD. In this sense, the term appears in the responses to the Reviewer's 2 comments. In the analyzes presented in this article, the threshold value was not used at all. Belonging to the group of people with a probable diagnosis of PTSD was determined using DSM-5 diagnostic criteria, which correspond to individual items of the PDS-5 questionnaire.

We feel, however, that Reviewer 2's use of the term "cut-off" refers to something else. The Reviewer points out that although individual items of the PDS-5 questionnaire refer to individual DSM-5 diagnostic criteria, the response scale is also used for each of these symptoms:

0 = Not at all

1 = Once a week or less/a little

2 = 2 is 3 times a week/somewhat

3 = 4 to 5 times a week/very much

4 = 6 or more times a week/severe

The Reviewer points out that perhaps if the value of 1 is already considered as an indicator of the occurrence of a single PTSD symptom, it is not enough, because the symptom may occur less than once a week and therefore does not interfere with functioning to such an extent that it justifies making a diagnosis of PTSD on the basis of such rare symptoms. His/her understanding of the term "cut-off" is comprehension at the level of a single item of the questionnaire, but in our study we followed the level of the probable PTSD overall score, as in DSM-5 diagnosis.

Therefore, this is a purely substantive issue and requires a diagnosis based on the literature and/or clinical practice. This diagnosis is based on the frequency of occurrence of symptoms that are significant for the subjects' functioning. But this topic is still the subject of ongoing discussion in the literature and there is no consensus in that topic (Foa et al., 2016).

And yes, in the discussion of the results, we commented on these issues, as well as the limitations mentioned by the reviewer. Nevertheless, we want to underline the fact that when including the subjects in the group of people diagnosed with probable PTSD, we took into account not only the occurrence of symptoms, but also their duration (criterion F) and - what is important - the impact on the functioning of the subjects (criterion G). In that way our results may be “protected” from overestimation of the level of probable PTSD diagnosis in our sample.

The authors have provided a table of descriptive statistics at the item level, but the point of criticism raised has not been sufficiently addressed. It would be helpful to include mean values, standard deviations, or median and quartiles, as well as extreme values, depending on the distribution, at both the criterion level and the total score level. This would enable readers to see how many people in the sample have very low values, which could be important for understanding the prevalence of subsyndromal PTSD. In addition, a visual representation by means of a scatter diagram would be even more helpful.

[Thank you for these remarks. We added Appendix B with descriptive statistics for PDS-5 scores and presented their distributions in Figure 1.]

To sum up, on behalf of my co-authors, I would like to take this opportunity to thank the Editor and Reviewers once again for their time and effort. We found all the comments very useful, and I believe they helped us improve the manuscript's quality. I sincerely appreciate the chance you gave us to revise and submit it to PLOS ONE.

---

## [Editor Report · Decision Letter 2]

14 Jun 2023

Exposure to Self-Reported Traumatic Events and Probable PTSD in a National Sample of Poles:

Why Does Poland’s PTSD Prevalence Differ from Other National Estimates?

PONE-D-23-05085R2

Dear Dr. Rzeszutek,

We’re pleased to inform you that your manuscript has been judged scientifically suitable for publication and will be formally accepted for publication once it meets all outstanding technical requirements.

Kind regards,

Inga Schalinski

Academic Editor

PLOS ONE
---

## [Editor Report · Acceptance letter]

29 Jun 2023

PONE-D-23-05085R2 

Exposure to Self-Reported Traumatic Events and Probable PTSD in a National Sample of Poles:
Why Does Poland’s PTSD Prevalence Differ from Other National Estimates? 

Dear Dr. Rzeszutek:

I'm pleased to inform you that your manuscript has been deemed suitable for publication in PLOS ONE. Congratulations! Your manuscript is now with our production department. 

Kind regards, 

on behalf of

Dr. Inga Schalinski 

Academic Editor

PLOS ONE